# Corm Rot of Saffron: Epidemiology and Management

Vishal Gupta [1,*], Akash Sharma [1], Pradeep Kumar Rai [1], Sushil Kumar Gupta [2], Brajeshwar Singh [3], Satish Kumar Sharma [4], Santosh Kumar Singh [5], Rafakat Hussain [5], Vijay Kumar Razdan [5], Devendra Kumar [5], Shazia Paswal [6], Vinod Pandit [7] and Rohit Sharma [8]

[1] Advanced Center for Horticulture Research, Sher-e-Kashmir University of Agricultural Sciences & Technology of Jammu, Udheywalla 180 018, India; akashskuastj@gmail.com (A.S.); pradeepr2000@gmail.com (P.K.R.)

[2] Division of Agroforestry, Faculty of Agriculture, Sher-e-Kashmir University of Agricultural Sciences & Technology of Jammu, Chatha 180 009, India; sushilgupta67@rediffmail.com

[3] Division of Microbiology, Faculty of Basic Sciences, Sher-e-Kashmir University of Agricultural Sciences & Technology of Jammu, Chatha 180 009, India; brajeshbhau@yahoo.co.in

[4] Seed Production Farm, Sher-e-Kashmir University of Agricultural Sciences & Technology of Jammu, Chatha 180 009, India; princesatish1@gmail.com

[5] Division of Plant Pathology, Faculty of Agriculture, Sher-e-Kashmir University of Agricultural Sciences & Technology of Jammu, Chatha 180 009, India; santosh_path17@rediffmail.com (S.K.S.); rafakathussain.rh@gmail.com (R.H.); vijayrazdan@gmail.com (V.K.R.); 11devrana@gmail.com (D.K.)

[6] School of Agriculture Sciences, Baddi University of Emerging Sciences and Technology Makhnumajra, Baddi 173 205, India; paswalshazia@gmail.com

[7] Centre for Agriculture and Bioscience International (CABI), New Delhi 110 012, India; v.pandit@cabi.org

[8] Regional Horticulture Research Sub-Station, Sher-e-Kashmir University of Agricultural Sciences & Technology of Jammu, Bhaderwah 182 222, India; rohit_agros@rediffmail.com

\* Correspondence: vishal94gupta@rediffmail.com

**Abstract:** Saffron, comprising of dried stigmas of the plant known as *Crocus sativus,* is one of the most important and scantly cultivated agricultural products. It has been used as a precious spice for the last at least 3500 years. Due to its numerous medicinal qualities and pharmacological applications, it is considered as a "golden condiment", and its demand and consumptions has risen over a period of time. Although efforts are continuously being made to enhance the productivity in the traditional areas and promote the cultivation of saffron in the newer areas, there are several constraints hindering these efforts. Prevalence of corm rot is one such limiting factor which results in the reduction in saffron production and decline in the area under its cultivation. The disease not only reduces the yield substantially, but also adversely affects the production of daughter corms. Complete understanding and knowledge about the disease is still lacking due to the inadequate information about its etiology and epidemiology. Moreover, due to the non-availability of resistant genotypes and lack of improved cultural practices, presently no effective and sustainable management strategies are available. This review article gives an overall account of the history and impact of saffron corm rot, its present status, yield losses caused by it, dynamics of the pathogens associated with the disease, their survival and dispersal, factors influencing disease intensity, epidemiology and sustainable management strategies. As comprehensive information on the disease is presently not available, an attempt has been made to review the current knowledge regarding corm rot of saffron. The information about the disease discussed here can eventually be beneficial for the growers, students, researchers, plant protection organizations, development departments, extension workers, policy makers, government agencies and public organizations.

**Keywords:** *Crocus sativus*; epidemiology; *Fusarium oxysporum*; distribution; corm rot; management

## 1. Introduction

Saffron is one of the most expensive dietary spices, which is obtained from the dried stigmas of *Crocus sativus* flowers. It is produced from the sterile autumn-flowering *Crocus*

*sativus* plant and comes from mauve-colored flowers having unique color, taste and fragrance [1]. The saffron plant is perennial herbaceous in nature, belonging to the family Iridaceae. Due to its properties as a functional food in the well-being of human beings against several diseases, it is considered as natural drug [2], therefore, its consumption is gaining importance. Carotenoids, glycosides, monoterpenes, aldehydes, anthocyanins, flavonoids, vitamins (especially riboflavin and thiamine), amino acids and proteins are the main constituents of saffron. Apart from these, starch, mineral matters, gum and many other chemical compounds are also present in it [3–6].

Saffron mainly grows in temperate and dry climatic conditions, however, its vegetative growth coincides with cold weather. Among the saffron producing countries, 88.80% contribution is from Iran, followed by India (5.80%), whereas, Greece, Afghanistan, Morocco, Italy, Spain, China and Azerbaijan contribute 1.90, 1.58, 0.68, 0.60, 0.26, 0.26 and 0.06%, respectively.

In spite of being a highly valuable spice with innumerable medicinal and culinary properties, there has been steady decline in the saffron production in various countries. The major bottle-neck for its successful cultivation is the non-availability of healthy corms in adequate quantity for use as the seeding material [7]. Moreover, its cultivation is highly labor intensive [8], and continuous genetic erosion [9], lack of mechanization, persistence with old cultivation practices and corm rot disease has contributed to the decline in the saffron production globally [10]. These obstacles have hindered the expansion of saffron production areas, therefore limiting its production at industrial scale, and making it unavailable to the consumers at a reasonable rate. All these factors have made saffron an important subject of scientific research regarding its cultivation technologies, potential medicinal properties and management of biotic stresses, which takes a heavy toll in the saffron production.

## 2. Socio-Economic Importance and Medicinal Properties of Saffron

Saffron, besides being a source for earning foreign trade, is strongly affiliated with the social, commercial and cultural ethics of the communities involved in its cultivation. *Crocus sativus* is the only plant species which produces apocarotenoids such as crocin ($C_{44}H_{64}O_{24}$), picrocrocin ($C_{16}H_{26}O_7$), safranal ($C_{10}H_{14}O$) and crocetin ($C_{20}H_{24}O_4$), which when glycosylated gets converted into crocin, in significant amounts. These compounds impart organoleptic properties to saffron, the dried stigmas of *Crocus* flower, making it the world's costliest spice. Crocin is responsible for the color, while picrocrocin attributes towards the bitter taste, and safranal provides the aroma.

Due to the presence of phytochemicals, saffron exhibits a broad spectrum of medicinal properties and has been used for the alleviation and treatment of several human ailments (Figure 1). It has been reported as hepato-protectant, reducing liver enlargement and aging of liver by acting up on liver antioxidants, peroxidation and detoxifification systems [11]. Saffron and its derivatives also possess cardio protective properties [12,13]. It inhibits pancreatic and gastric lipase activity and increases fecal excretion of rat by reducing blood lipids [14]. It possesses the anti-catarrhal activity and is used for ophthalmological treatments such as diabetic glaucoma and also age-related macular degeneration [15]. Saffron extracts exhibit anti-hyperglycemic property in the alloxan-induced diabetic rats [16]. Its extract also reduces plasma cholesterol and triglyceride in rats, which might be due to the scavenging activity of crocin and safranal [17]. Due to strong antioxidant and radical scavenging properties, crocetin and crocin are also responsible for anti-inflammatory activity [18] and even act as anti-arthritic agents. Aqueous and ethanolic extracts of saffron also show anticonvulsant activity [19]. Mice treated with it exhibited a decrease in depressive condition [20,21]. The saffron extracts also have inhibitory effect on the *Streptococcus mutans*, *Lactobacillus* and *Candida albicans*, responsible for dental caries [22].

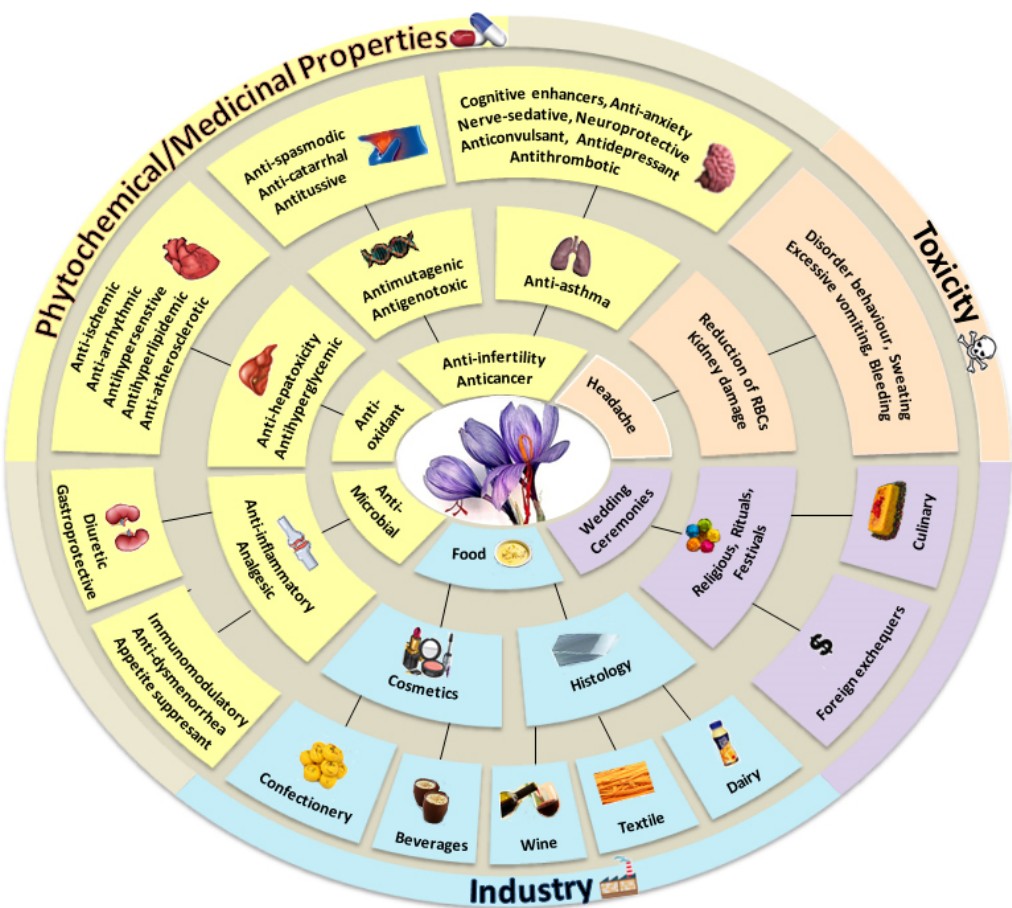

**Figure 1.** Potential uses of saffron.

Treatment with ethanolic extract of saffron, was seen to have antiproliferative effect by significantly reducing the longevity of malignant carcinomic human alveolar basal epithelial cells [23]. Saffron derivatives having higher anti-carcinogenic potential has attributes of increased apoptosis, reduction in blood vessels [24], inhibiting RNA and DNA synthesis [25] and regulating lipid peroxidation and antioxidant activity [26]. Apart from the inhibition of cell proliferation, enhancement of cell differentiation, modulation of tumor metabolism, cell growth and immune modulation are the other mechanisms exhibited by the chemo-preventive activity of saffron [27].

Aqueous extracts of saffron reduces different pro-inflammatory factors that are involved in the pathogenesis during anxiety [28]. Saffron suspension effectively stimulates humoral and cell-mediated immunity [29]. Studies have also shown the immunological attributes of saffron and its constituents. Saffron flavonoids, because of their potent antioxidant effects [30] are an effective treatment for neurological diseases including Alzheimer [31–34]. Crocin has exhibited protective effect against Parkinson-like symptoms [35]. Regular consumption of saffron is reported to ameliorate anxiety and depression in adult patients [36]. However, its consumption, during last trimester of gestation, results in preterm delivery due to the increased uterine contractions in female mice [37]. In Chinese medicines, saffron is recommended for labor pain, menorrhagia and postpartum hemorrhage [38].

## 3. Center of Origin and Geographical Distribution of Saffron Crop

The center of origin of saffron crop is not clear, as earlier Middle East (Asia Minor, Turkestan and Iran) was considered as its center of origin [39], whereas, later on, Crete (Eastern Greece) was proposed as its center of origin [40], and Mesopotamia has also been regarded as the place of its origin [41]. Based on the scientific evidence, the Mediterranean

region is also assumed as one of the probable regions of the origin of saffron, and in the East, Turkey, Iran and India, with a history of saffron cultivation dating back to thousands of years, are also regarded as other sites of its origin [42]. Presently, the majority of the saffron growing areas are confined to the section of the Mediterranean region in the West and Iran, Turkey and Jammu and Kashmir (India), in the East [43].

Saffron, once neglected by researchers and farmers, has attained special attention during the last couple of decades, after realizing its therapeutic effects on human health. Nevertheless, considering the fact that saffron cultivation has significant economic and social benefits to growers, newer areas are being explored by the public and private sectors, as well as by the national and international agencies, in order to enhance saffron production and productivity (Figure 2). Presently the main area of its cultivation is Khorasan Razavi (82,712 ha), South Khorasan (15,754 ha), North Khorasan (5260 ha) and other provinces (1544 ha) of Iran [44]. Globally, India has the second largest area under saffron cultivation. In India, it is mainly cultivated in the four districts of Union Territory of Jammu and Kashmir namely, Pulwama, Budgam, Srinagar and Kishtwar, with 3200, 300, 165 and 120 ha under cultivation, respectively. However, saffron productivity in India is very low (2.0–2.5 kg/ha), as compared to the other saffron producing countries.

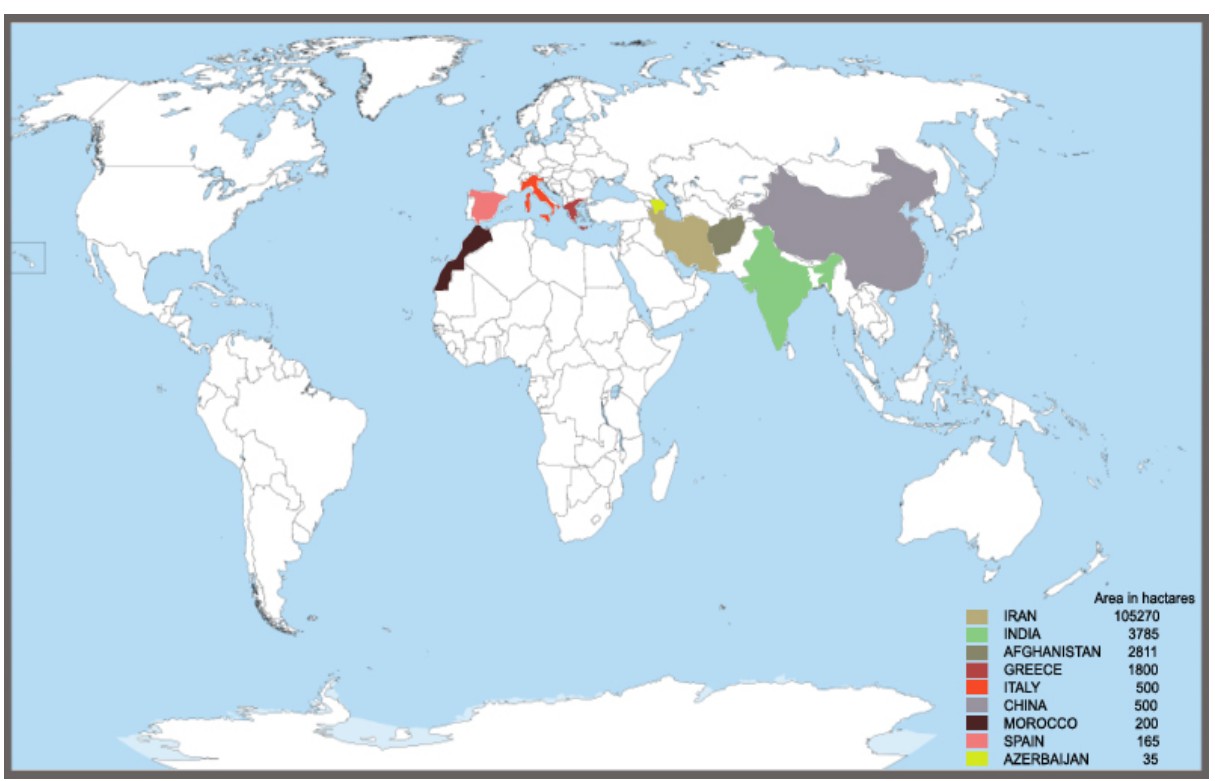

**Figure 2.** Major saffron growing countries of the world.

In Spain, the main saffron producing area is Castile-La Mancha, however, from 1997 to 2016 the area under its cultivation reduced from 841 to 165 ha, whereas its productivity increased from 8.82 to 14.00 kg/ha. In Afghanistan, saffron cultivation initiated in early 1990s, and due to the joint efforts of Government and development partners, it is being cultivated on 2811 ha. In Greece, the major area under saffron cultivation (1800 ha) is Kozani in Macedonia [6]. In Italy, it is mainly cultivated in L'Aquila, Sardinia, Tuscany and Umbria, over an area of 500 ha. In Morocco saffron cultivation is limited to Taliouine and Taznakht regions, over an area of 200 ha. In Turkey, saffron was cultivated in Safranbolu, Istanbul, Izmir, Tokat, Adana and Sanliurfa regions during the Ottoman empire, but presently it is cultivated on only 2 ha [45]. In Azerbaijan, saffron is cultivated on the Apsheron peninsula near the city of Baku, where before 1917, it was cultivated on over 150 ha, but due to

several constraints its cultivation is now restricted to 35 ha only. In China, saffron was introduced in 1986 and is mainly grown in Zhejiang, Jiangsu and Shanghai provinces, over an area of 500 ha. The countries having a maximum share in the global export of saffron are Iran (USD 114M), Spain (USD 63.2M), Afghanistan (USD 18.4M), Greece (USD 6.97M) and France (USD 6.42M), whereas the top importers are Spain (USD 54.7M), Hong Kong (USD 20.8M), the United States (USD 18M), Italy (USD 16.7M) and India (USD 16.6M) [46].

## 4. Corm Rot of Saffron

Globally, the area under saffron cultivation has shown significant decline due to several biotic and abiotic factors. Among the various biotic constraints, corm rot has been recognized as a major limiting factor for successful cultivation of saffron in both traditional and non-traditional areas. The disease is also known as dry rot, brown rot, violet root rot, copper web, basal rot, corm rot, death blight and yellows. Due to the association of several pathogens and secondary saprophytes with the rotting of the corm, the disease is considered as corm rot complex. The disease was first reported to be caused by *Rhizoctonia crocorum* by Duttamel in 1728, and it was responsible for forcing the growers to abandon the cultivation of saffron in France and England. In Japan, the disease was first encountered in 1909 in Fuchu Tokyo-Fu and Kozu, Kanagawa-Kenin, and subsequently in 1918 in Kanagawa-Ken Agricultural Experiment Station, and in 1919 in Okazaki. Occurrence of saffron corm rot complex was reported as dry rot in Kyoto and Hyogo in Japan [47,48]. The disease has also been reported in Spain and France [49], Spain [50,51], Italy [52–54], Greece [55], Morocco [56], Scotland [57], Iran [58,59], Netherlands [60], China [61,62], Almora (India) [63], Kishtwar (India) [64–66] and Pampore (India) [67,68]. In Central Italy (L'Aquila), poor growth and wilting in the saffron fields due the corm rot caused serious economic losses to the farmers during 1988 and 1989 [69], and led to significant reduction in the yield, so much so that in some cases a complete loss of the saffron crop was recorded [70]. The disease has since been reported globally causing substantial yield losses, mainly due to the sequential cultivation and planting of the infected corms. Presently, corm rot complex is prevalent in moderate to severe form in all the saffron growing areas of the world, resulting in significant reduction in the production and productivity of saffron.

In Jammu and Kashmir (India), the disease is primarily noticed during flowering (October–November) and grubbing period (May–July). In twenty-five commercial fields of Kishtwar District of Jammu and Kashmir, 70 to 80% of saffron crop was found infected with corm rot [71]; at Berwar (Kishtwar), the disease incidence ranged from 30 to 40% [64]. Disease incidence and severity in the range of 42.67–59.33% and 17.78–35.00%, respectively, have been recorded in Kishtwar [66]. In Kashmir, every saffron growing field was found infested with corm rot [67], with infestation of 70–80% [72]. Moreover, corm rot incidence ranging from 4.00 to 40.00% and 4.66 to 42.00%, and the intensity ranging from 0.8 to 16.93% and 1.33 to 17.43% in 1999 and 2000, respectively, was observed in Kashmir [73]. The disease is considered as a major threat to saffron industry in India as 21% of the saffron area in Pampore of Kashmir is highly infested with the disease [74]. Hundred per cent disease incidence with the severity of 6–46%, resulting in reduced plant growth and yield of saffron in Kashmir has been reported [75].

### 4.1. Symptoms

Due to the association of various plant pathogenic microorganisms with corm rot complex of saffron, varied symptoms were exhibited by the infected host. Corm rot due to *Fusarium* spp. is characterized by yellowing, drooping and wilting of the shoots during flowering period [76], resulting in the death of foliage. On the upper surface, initially the infected corms show small specks with chlorotic haloes, which later on spread to the whole corm and eventually the entire corm turns into dark powdery mass and decay. As the disease progresses, brown to dark brown, sunken irregular patches below the corm scales are observed. In certain cases, dark lesions beneath the outer tunic layer of the corms appear along with the blue-green mold due to *Penicillium* spp. [69]. At a later

stage of disease development, white fungal mycelium may appear on the rotten corms. Sclerotia formation has also been observed on the diseased corms due to the infection caused by *Sclerotium rolfsii* and *Rhizoctonia* spp. [65]. Infected plants die early, resulting in the reduction in size and number of daughter corms and flowers (Figure 3). This leads to the reduction in flowering period and subsequently manifest into low yield and poor quality of saffron [77]. Pathogenic fungi infect the corms by penetrating through the protective sheaths, subsequently converting the white-colored corm surface to yellow and ultimately to black, resulting in the rotting and death of the invaded corms [49]. Under moist conditions, plants infected with corm rot rapidly turn yellow and then brown, but under dry conditions, the infection progresses very gradually and ultimately causes drying of the infected parts. In newly infested fields, the disease first occurs in small patches that gradually enlarge with each passing year until the whole field is infested. Due to the planting of diseased corms, in the following year, all the leaves coming out from these corms appear yellow, and the severely infected corms altogether fail to germinate or sprout. The corm rot symptom initiates with the corm germination and root initiation stages and continues until the storage of corms.

### 4.2. Pathogens Associated with Corm Rot

Saffron corm, as a subterranean organ, is more likely prone to the attack of soil-borne fungi and bacteria. Apart from the microorganisms present in soil, environmental conditions, soil chemistry, irrigation and use of fertilizers have major effects on the production of saffron and corm quality. Different groups of plant pathogens have been found to be responsible for saffron corm rot complex. The microbial presence and interactions among them, in the *Crocus sativus* rhizosphere and cormosphere, are not only organ specific but are also affected by various plant growth stages. The prevailing weather parameters also have their impact upon the seasonal variations in the dynamics of microbial community. To have the knowledge of such interactions in the rhizosphere and cormosphere is very helpful in understanding the changes occurring in the microbial community over the time, in relation to the environmental variations, and its effect on plant health.

Various plant pathogens causing corm rot disease, either singly or in combinations, inflict heavy yield losses in saffron. Microorganisms, such as *Bacillus croci*, *Fusarium bulbigenum* var. *blasticola*, *Sclerotinia gladioli* and *Fusarium oxysporum* f. sp. *gladioli* have been identified to be associated with the rotting of saffron corms from Japan [47,48,78]. From the infected corms, in Italy, *Penicillium cyclopium*, *F. oxysporum* f. sp. *gladioli* and *Burkholderia gladioli* have been observed [52,54,76]. *Penicillium corymbiferum* was reported from Scotland, Italy and China [57,79,80]. From Spain, *Sclerotinia bulborum* and *Rhizoctonia crocorum* have been reported as the causal organisms of corm rot of saffron [50]. *F. oxysporum* f. sp. *gladioli*, *Macrophomina phaseolina*, *F. oxysporum*, *F. solani* and *Sclerotium rolfsii* have been reported from India [63–65,71,81]. *Pseudomonas gladioli* has been reported to be associated with saffron corm rot from China [61]. *F. oxysporum*, *F. culmorum*, *F. roseum*, *Fusarium* sp., *Aspergillus* , *A. niger*, *Rhizopus oryzae*, *Penicillium* sp. and *Rhizoctonia violacea* have been observed in Morocco [56,82]. From Iran, *F. oxysporum*, *A. niger*, *P. digitatum*, *R. stolonifera*, *Aspergillus terreus*, *A. flavus*, *A. flavipes* and *A. niger* have been encountered from the infected corms [59,83,84]. Severe infection of *P. corymbiferum*, reducing corm yield by more than 20% has been reported [57]. Fungal pathogens like *Pythium* spp., *Rhizoctonia* spp., *Phoma* spp., *M. phaseolina*, *Penicillium* spp., *F. solani*, *R. crocorum* and *Phoma crocophi* have also been found associated with the corm dry rot [49,53,64,68,85].

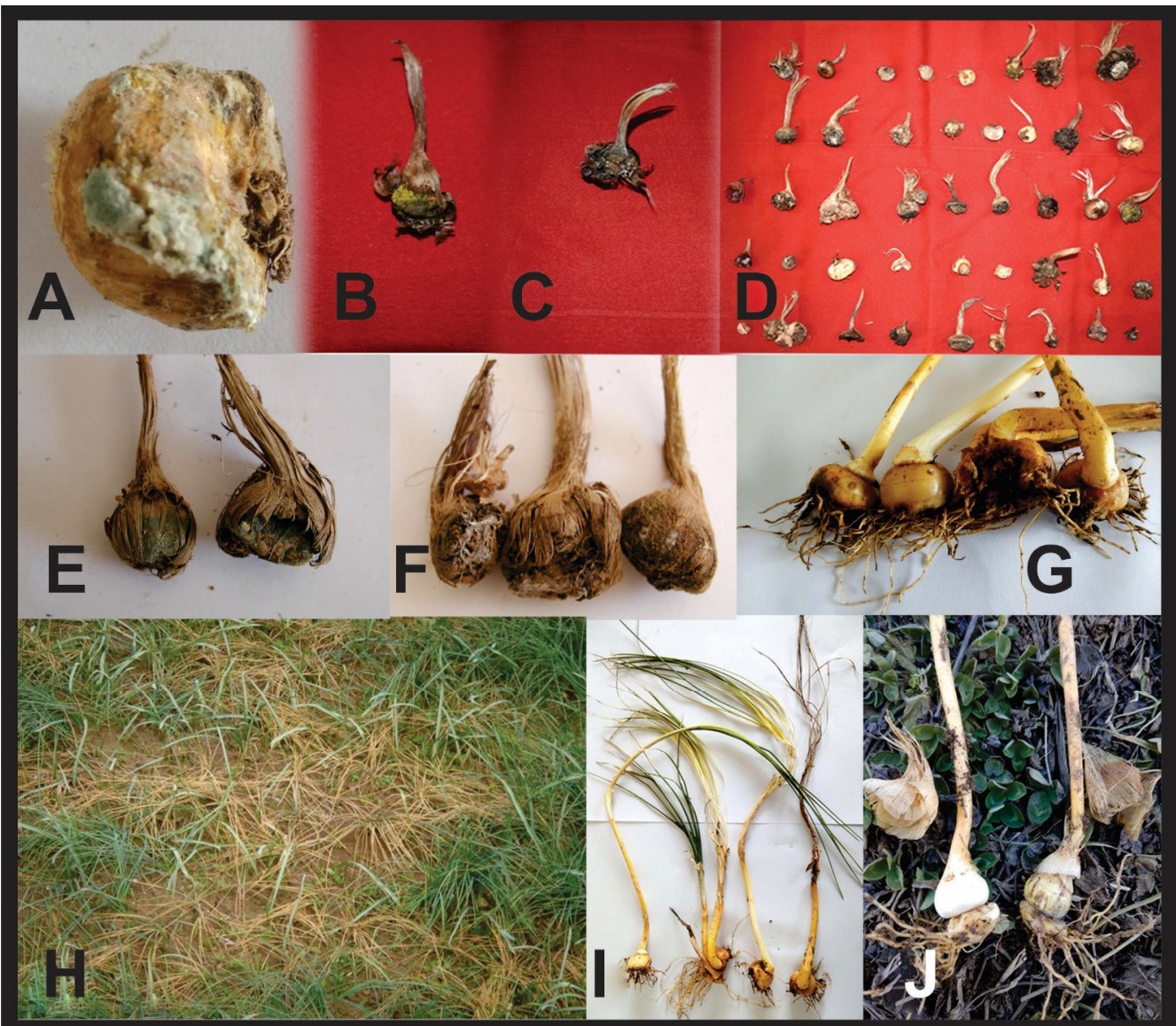

**Figure 3.** Symptoms of saffron corm rot. (**A**) Growth of *Penicillium* on infected corm; (**B**) Rotting of saffron corms by *Botrytis* spp.; (**C**) Rotting of saffron corms by *Rhizoctonia* spp.; (**D**) Corm rot complex of saffron; (**E**) Rotting of saffron corms by *Penicillium* spp.; (**F**) Rotting of saffron corms by *Sclerotium* spp.; (**G**) Saffron corms infected by *Fusarium* spp.; (**H**) Diseased saffron plants in the field; (**I**) Infected saffron Plants; .; (**J**) Tunic layer of saffron corms infected by *Fusarium* spp.

Presence of plant parasitic nematodes are known to aggravate the soil-borne diseases caused by the *Fusarium* spp. They act as bio-predisposing agents in encouraging infection of plants by creating wounds [86], modifying the host (localized and systemic modifications), influencing root exudations and altering host immunity [87]. Nematodes cause lesions on the root, subsequently weakening the plant and easing the penetration of the pathogen [88], leading to the increase in the population of *F. oxysporum* [89]. *Tylenchus, Helicotylenchus, Pratylenchus, Hirschmaniella* and *Psilenchus* are the nematodes genera reported from the saffron fields of Kashmir are [90], and 11 genera have been found in Spain [91]. *Aphelenchoides, Ditylenchus, Pratylenchus* and *Helicotylenchus* were reported to parasitize the saffron crop in Morocco [92]. *Pratylenchus thornei, Helicotylenchus vulgaris, Aphelenchus avenae, Xiphinema* sp., *Tylenchus* sp., *Tylenchorhynchus* sp. and *Hemicriconemoides* sp. were found infesting saffron soil with per cent infestation of 8.86, 16.64, 5.87, 14.64, 13.00, 10.70 and 3.26, respectively [93]. Fifteen plant parasitic nematode genera belonging to twelve different families were identified, in Morocco, among these *Ditylenchus, Aphelenchoides, Pratylenchus*

and *Helicotylenchus*, were mainly responsible for the reduction in saffron production. Soil texture, humidity, rainfall and temperature were the major factors influencing the presence and the population of nematode genera. *Ditylenchus*, *Helicotylenchus* and *Paratylenchus* were reported mainly from the humid and silty soils, whereas the presence of *Aphelenchoides*, *Tylenchus*, *Tylenchorynchus* and *Dorylaimus* was significantly high in locations with regular rainfall and clay soils [92].

### 4.3. Mycoflora Associated with Corm

Population of fungi such as *Ilyonectria macrodidyma*, *Mortierella humilis* and *Chaetomiaceae* was found to be significantly higher in soil samples supporting good corm growth. The fungi such as *Nectriaceae*, *Penicillium*, *Aspergillus* and *Saccharomycetales*, mainly affected corm germination and blooming, whereas corms that could germinate but did not bloom were infected by *Serratia* and *Penicillium chrysogenum* [94]. Population of *Trichocomaceae* and *Talaromyces* was high in the infected corms, whereas corms that were not able to germinate had higher population of *Aspergillaceae. Penicillium* and Dothideomycetes were observed in corms that could germinate but not bloom, and in corms that could germinate and produce only two flowers per corm, respectively. To catalogue fungal diversity in the rhizosphere and cormosphere of *Crocus sativus*, pyrosequencing data analysis revealed *Zygomycota* as the dominant fungal phylum in the rhizosphere, whereas *Basidiomycota* was dominant in cormosphere during flowering stage. The phylum dominant in the cormosphere at dormant stage was rarely found at flowering stage and vice-versa. Among various genera, *Rhizopus* was dominant in dormant stage, though it was rare in flowering stage. *Ascomycota* dominated the bulk soil during both the stages. However, *F. oxysporum* causing corm rot in *C. sativus*, has been found to be present during both the stages with slightly higher abundance in roots [95].

Like other ornamental plants belonging to the family Iridaceae, such as *Crocus*, *Gladiolus*, *Iris* and *Ixia*, saffron is also susceptible to *Fusarium oxysporum* f. sp. *gladioli*. *F. oxysporum* f. sp. *gladioli*, *F. oxysporum*, *F. solani*, *Rhizoctonia* sp., *Aspergillus* sp., *Penicillium* sp. and *Macrophomina* sp. have been isolated from the diseased saffron corms, with *F. oxysporum* f. sp. *gladioli* being the predominant pathogen [66,75]. A basidiomycetous latent pathogen of saffron, identified as *Porostereum spadiceum*, has also been reported, which was capable of producing phytotoxic compounds resulting in corm rot under in vivo and field conditions, with a disease severity index of 0.7 and 0.5, respectively [96]. In order to understand the formae speciales of *F. oxysporum*, inciting corm rot in saffron and its host range, pathogenicity tests of formae speciales colonizing saffron were conducted on other members of Iridaceae, such as *Crocus vernus* (ornamental crocus), gladiolus and narcissus. It was observed that *F. oxysporum* formae speciales *iridiacearum*, *croci* and *saffrani* were pathogenic to all the crops tested, *Crocus* species and saffron, respectively. Therefore, creation of a new formae speciales, *saffrani*, which showed pathogenicity only on saffron corms was suggested [77].

Endophytes, such as fungi and bacteria, are present between the living plant cells, and have relationship with the plant varying from symbiotic to bordering on pathogenic. From the corms of *Crocus sativus*, 294 fungal endophytes have been isolated in Jammu and Kashmir. The diversity of endophytes, with *Phialophora* mustea and *Cadophora malorum* being the dominant ones, has been observed to be higher during the dormant stage than at the vegetative stage. Endophytes that produce indole acetic acid generally possess antimicrobial properties, that are helpful for the host in evading the pathogens. However, thirteen endophytic taxa have been identified to cause corm rot in the host with different levels of severity under in vitro as well as in vivo conditions [97]. Fungal endophyte, *Mortierell aalpina* CS10E4, was found to significantly improve the morphological and physiological traits in *Crocus* plants, by way of enhancing the total biomass and corm size, stigma biomass, number of apical sprouting buds and number of adventitious roots. This strain also enhanced the production of apocarotenoids and increased tolerance towards corm rot by releasing arachidonic acid which acts as conserved defense signal and induces the production of jasmonic acid. However, some endophytic strains of *Alternaria alternata*,

*Aspergillus pseudodeflectus* and *Penicillium pinophilum* have been reported to cause extensive rotting in saffron corms, while those of *Fusarium oxysporum*, *Penicillium canescens*, *Talaromyces cellulolyticus*, *Acremonium alternatum*, *Botrytis fabiopsis*, *Porostereum* sp., *Epicoc cumnigrum*, *Paecilomyces marquandii*, *Talaromyces pinophilus* and *Talaromyces verruculosus* cause moderate rotting [97,98].

*4.4. Disease Cycle and Epidemiology*

Saffron is a perennial crop and the planting cycle varies from 5 to 6 years. Planting is generally carried out without proper phytosanitary considerations; hence the infected and/or infested corms act as a source of primary inoculum of the disease. Prevalence of large number of pathogens with corm rot complex of saffron intricate the epidemiology of the disease. Detailed disease cycle and epidemiological studies of specific pathogens associated with the disease are yet to be explored. Nevertheless, it is essential to know the occurrence, synergistic activity among the pathogens in aggravating the disease, infection process and relation of abiotic factors with the disease initiation and development. *Fusarium oxysporum*, the most dominant pathogen causing the corm rot, survives in diseased corms and soil by means of mycelium, chlamydospores, macroconidia and microconidia [99,100]. Thick walled chlamydospores are resistant to desiccation, and therefore resist unfavorable environmental conditions and may survive in the soil for more than 20 years [101–103]. The ability of *F. oxysporum* to grow saprophytically and colonize the plant debris accelerates chlamydospore production, thus increasing persistence of pathogen in the soil [104]. Once the corms are sown in the soil and on the onset of favorable environmental conditions, the fungal chlamydospores/spores germinate and penetrate into the corm tissue by means of germ-tubes, thus setting in the infection process resulting in the production of typical symptoms. Sclerotia, formed by fungi like *Rhizoctonia* and *Sclerotium*, not only help pathogens to overcome the unfavorable environmental conditions, but also serve as the major source of infection [105,106]. Sclerotia also germinate, once the environmental conditions are favorable, and give rise to mycelium which cause infection of the corms present in the soil.

The perennial monocropping system of saffron cultivation encourages continuous disease cycle and also increases inoculum build-up in the soil. As the corm rot is polycyclic in nature, and there is continuous presence of host in the field, even the small amount of inoculum can result in devastating loss and total failure of the crop. Higher corm rot intensity has been reported in arid soils compared to that in humid soils [47]. Well-drained clay calcareous soils with a fairly loose texture are good for saffron cultivation, as it supports easy root penetration. Under waterlogged conditions, the corms rot intensity increases and therefore proper drainage system in the field is important for the productivity of saffron [107,108]. Besides climate and soil, planting time, seed/corm rate, planting depth, corm size/weight, crop density, nutrient management, weed management, harvest and post-harvest management also influence saffron quality and quantity [109].

Active dispersal of the pathogen takes place through roots of infected corms. Once the infected corms decay in the soil, the spores are set free, and upon germination cause fresh infections to the newly sown corms or the daughter corms. Production of daughter corms in the proximity of infected corms results in the formation of interconnected mats of infected roots which accelerates the disease intensity within the field. The dispersal of the pathogen propagules, associated with corm rot of saffron, occurs rapidly within the field, locality, country or even continent by their passive movement. Long distance dispersal of corm rot pathogens is mainly due to the exchange of propagating material (corm), while short distance dispersal may be associated to both anthropogenic (contaminated farm implements, tools, labor, clothes and footwear) and natural factors, such as water, movement of animals or soil infested with the pathogen and air [110]. Since globally the saffron crop has mainly been established by invaders and visitors [58], pathogens too may have dispersed into new regions or field through import or exchange of infected corms [111] or even by new local emergence [112]. Apparently symptomless and infected daughter corms act as a potential inoculum reservoir for the dispersal of corm rot disease.

The lack of certification mechanism while exchanging the propagating material (corm), helps in the dissemination of the disease at local and regional scales. Planting material as such is considered as one of the most important factors for the dissemination of Fusarium wilt propagules [113–115].

## 5. Management

### 5.1. Resistance

Since the saffron plant has a male-sterile triploid lineage, therefore, ever since its origin it is propagated only through vegetative means by corms, thus limiting the chance of its genetic improvement. In order to protect its bio-diversity, 220 accessions from 15 countries have been collected at the Bank of Plant Germplasm of Cuenca, Spain [116]. Nevertheless, compilation of traits related to the morphological, molecular, phonological, agronomical and biochemical aspects related to the genus *Crocus* are needed for the improvement of saffron crop. Moreover, knowledge regarding its susceptibility to stress factors, resistance towards the diseases and abiotic factors are also needed for the crop improvement. Presently, tissue culture-derived plantlets are the most reliable source of pathogen-free material and can be explored as the planting material.

### 5.2. Exclusion

Exclusion is the primarily measure to manage the disease and avoid its establishment in pathogen-free regions. To avoid the entry of the pathogens in newer areas, it is essential that the planting material is properly screened for the presence of any inoculum. Quick and reliable diagnostic tools for specific detection of *Fusarium oxysporum gladioli* infection in propagation material can be explored by vegetative compatibility grouping [116]. With the advancement in DNA-based diagnostic tests which have numerous advantages, such as non-culturing of the target pathogens, ability to identify and detect an individual strain from the mixed sample, quantification of the target pathogen, detection of multiple pathogens, less time consuming and cost effective, help in the identification and enumeration of the pathogenic strains of corm rot of saffron [117].

### 5.3. Cultural Practices

Modification, improvement and standardization of existing practices of saffron cultivation can greatly help in reducing the incidence of corm rot. Grading and sorting of saffron corms before sowing must be the pre-requisite for the better crop health and yield. Nutrient reserves present in the mother corms have an effect on saffron growth, particularly during early growth stages. The size of corm generally varies from 1 to 20 g, and is directly related to flowering and number of flowers produced [118,119]. Small sized corms (<6 g), either fail to flower or have limited potential of flowering [120]. Optimum corm size (8 g) must be preferred for rising of saffron crops, as it directly enhances the yield and production of daughter corms. However, the undersized and small corms must be sorted out and selected for raising the nursery of saffron corms, which can ensure the enhancement of corm size and better production of daughter corms for the subsequent crops. Planting of corms without sorting is the main reason for spread of corm rot inoculums.

It is necessary to take the utmost care during storage, transportation and handling of saffron corms to avoid any injury, damage or wound that would predispose them to corm rot complex. Resource-poor farmers, lacking proper storage facility, usually tend to predispose the corms to such injuries and pathogenic infections. Storing the corms in heaps or stacking must be avoided, and the corms must be placed in layers for proper air circulation and to reduce the chances of injury to the corms. During packing and transportation of corms, adequate aeration is essential. Any type of injury, wound, or damage not only reduces the vigor of the corms but also aggravates the chances of infection in the seed lot. To avoid long storage periods, corms must be distributed immediately after transportation for sowing purposes, as poor storage conditions result in rotting of corms. Proper corm rate, density, depth, spacing and planting method significantly improve the

health of the crop, and also reduce the chances of corm rot complex. It has been observed that maximum and minimum number of flowers, dry weight of flowers and stigma were recorded in the plants where corms were sown at 10 and 5 cm depth, respectively [121]. It has also been seen that the yield, number of daughter corms, phosphorus content, flower numbers, fresh flowers and dry stigma yields significantly decreased by irrigating at 50% of saffron water requirement (SWR) as compared to 75 or 100% SWR [122]. Larger mother corms (>8 g) result not only in greater nitrogen content in daughter corms and the whole plant, but also in better acquisition and use efficiency of N and P [123]. Saffron cultivation is mainly restricted to the regions with cold and dry winters, therefore the damage induced by freezing is one of the most adverse factors increasing electrolyte leakage and reduced corm growth. It has been observed that mother corms obtained from 4-year-old farms and sown at 20 cm soil depth effectively overcame the adverse effects of freezing injury in saffron [124].

### 5.4. Crop Rotation

Saffron is a perennial crop and its cultivation is invariably restricted to the same piece of land, thereby promoting the chances of inoculum build up in the soil and enhancing the corm rot incidence. Continuous monocropping at the same site tends to gradually deplete certain nutrients in the soil, and also encourage a highly competitive pest and weed community in that location. Furthermore, crop rotation is the system of growing different types of crops successively at the same site across the growing seasons to maintain soil fertility and to control weeds, diseases and pests. A properly thought-out crop rotation can not only improve soil structure and matter, but also reduce the pathogen and pest population in the field. Thereby, it can act as an effective management practice against soil-borne pathogens in semi-perennial crops like saffron, and be a crucial factor for the maintenance of soil fertility. It also significantly lowers the inoculum load in the soil by helping to create a suppressive environment and reducing or eliminating weed hosts in the field [110]. Though the information available regarding the effect of previous crop on saffron yield and quality is scanty, a gap of 3 to 8 years should be maintained before it is cultivated on the same piece of land [10]. Decrease in the production of stigma and an increase in weed population were recorded when saffron was continuously cultivated in the same field. Notably, saffron cultivation was found to be more remunerative if it was rotated with legumes and wheat [125]. Out of two previous crops (faba bean and saffron) and fallow, faba bean contributed towards highest flower number and stigma yield in saffron [126]. Linseed, maize and oat have been introduced in the cropping system of saffron in Jammu and Kashmir [127].

### 5.5. Chemical Management

Application of fungicides is popular, readily available, easy to handle over larger areas, and an effective method of managing diseases in agricultural crops. Under in vitro conditions, complete mycelial inhibition of *F. oxysporum* and *F. solani* isolated from the infected saffron corms, using carbendazim has been recorded [128]. Carbendazim was also reported to be highly effective in checking the mycelial growth of *F. oxysporum* f. sp. *gladioli* isolated from diseased corms [71]. Dipping *Crocus* corms infected with *Penicillium corymbiferum*, in benomyl, captafol, tnancozeb or thiram solutions has been found effective under laboratory conditions [57]. Corm treatment with carbendazim not only enhances the average yield of saffron, but the rotting of corms is also reduced, resulting in better saffron yield [129]. Drenching of root system of diseased saffron plants with Brassicol at 1.5–3.0 g ai/m$^2$, on the appearance of corm rot symptoms successfully manages the disease [58] and Prochloraz treatment before planting produces healthy plants and corms [130]. Dipping of corms with difolatan (80% captafol), Bavistin (carbendazim) or benlate (benomyl) each at 0.2% for 20 min before planting successfully manages corm rot of saffron caused by *Fusarium oxysporum* f. sp. *gladioli* [63]. Carbendazim and thiobendazole at 0.2% as a dip or drench is reported to give complete control of corm rot of saffron [81]. Overnight

dipping in the suspension of carbendazim or myclobutanil at 0.2% each, reduce the corm rot severity by 7.4 and 5.2%, respectively, compared to 46.7% in control [131]. However, growing concerns about the deleterious impacts of synthetic chemicals on the environment as well as human beings along with their residues in the produce has made it mandatory to explore the alternative methods for the management of soil-borne diseases which are eco-friendly and sustainable.

### 5.6. Biological Control and Plant Growth Promotion

Biological control has emerged as an alternative to chemical management because of the former being a sustainable approach having no residual activity or non-phytotoxic properties. Soil application of talc-based formulation of *Trichoderma viride* at 2.5 kg/ha and subsequently corm treatment with carbendazim at 0.2% has been found effective in reducing the incidence of corm rot and simultaneously ensuring higher germination [71]. *T. viride*, when applied as corm treatment or into soil through FYM, is reported to minimize the corm rot severity [132]. *T. viride* as a corm dresser, in combination with direct application of vermicompost and neem cake to the soil, managed saffron corm rot to the extent of 98%. The combined treatment of *Trichoderma harzianum*, *Azospirillum* spp. and mustard cake, resulted in maximum corm yield, number of daughter corms/mother corm, fresh flower weight and saffron yield [133]. *Bacillus* spp., having the advantage of producing heat- and desiccation-resistant spores, significantly reduce the incidence of saffron corm rot under pot conditions [74], which may be attributed to the competition for energy, food and ecological niche or substrate, production of inhibitory allelo-chemicals and induction of systemic resistance [134]. Corm treatment with *B. subtilis* significantly reduces the days of sprouting and flower emergence along with causing an increase in number of flowers [66]. Dipping of saffron corms in *B. subtilis* FZB24 suspension for 15 min before sowing significantly increased leaf length, flowers per corm, total stigma biomass and decreased the number of days for corm sprouting. The quantity of crocin, picrocrocin and safranal is also significantly increased when the saffron fields are drenched with the *B. subtilis* FZB24 [135]. It has been observed that the population of *F. oxysporum* f. sp. *gladioli* drastically reduced after 30, 60 and 90 days of sowing of saffron crop with maximum reductions of 58.33, 66.66 and 72.50%, respectively, by soil solarization. It was followed by the treatments with *T. viride*, *T. harzianum* (each at $1 \times 10^7$ cfu/mL), *Pseudomonas fluorescens*, *Bacillus subtilis* (each at $1 \times 10^9$ cfu/mL), Vesicular Arbuscular Mycorrhizae (1 kg/acre) and carbendazim 50 WP (0.2 %) [66].

The inoculation of arbuscular mycorrhizal fungi (AMF) significantly increases the fresh and dry weight of saffron plant biomass along with the photosynthetic pigments and soluble sugar content [136]. AMF inoculation results in significantly higher initiation and stimulation in the production of new corms and cormlets, and also enhances the growth characteristics of fresh and dry matter in saffron. With the inoculation of AMF, faster corm initiation after bud and leaf emergence takes place as compared to non-inoculation [137]. It has been observed that the application of microbial inoculants (*Azospirilium* spp. + VAM + *Pseudomonas* spp.) increase the leaf number, plant height and number of flowers in saffron crop [138]. Application of plant growth promoting fungi (PGPF) and plant growth promoting bacteria (PGPB) are widely used against an array of different crops for combating biotic and abiotic stresses for sustainable agriculture. These beneficial microbes promote plant growth by diverse mechanisms such as production of useful metabolites and signals, phytohormones, bio-fertilization, phyto-stimulation, antibiotic resistance, bio-remediation, rhizosphere engineering, lytic enzymes, increased resistance to osmotic stress and other abiotic factors [139–144].

*Mortierella alpina* CS10E4, a fungal endophyte isolated at dormant and vegetative stages of *Crocus* life cycle, is seen to reduce the mycelial growth of *F. oxysporum* by 53.5% and disease severity of corm rot by 50.0%, due to the production of arachidonic acid in endophyte-treated plants. The endophyte significantly increases corm size and biomass and also the number of apical buds sprouting per corms, which have a direct relationship

with flower formation due to the production of significant amounts of indole acetic acid, siderophore and chlorophyll content in the leaves. Saffron plants inoculated with *M. alpina* (CS10E4) also exhibited about two- and four-fold increases in crocin and safranal contents, respectively, as compared to the control ones [98]. In tissue culture studies, *Pseudomonas* sp., *Bacillus subtilis* and *Pantoea* sp. were observed to significantly increase the weight of cormlets. Similarly, proliferation of cormlets was also significantly improved with *Pantoea* sp., *B. subtilis,* and *Pseudomonas* sp. on MS liquid medium. Simultaneous cultivation of saffron with *Acinetobacterhae molyticus*, *Accintobacterl woffii* and *Pantoea* sp. resulted in 100% germination of cormlets [145]. *Curtobacterium herbarum* Cs10 improves the flower numbers and significantly enhances the length of the stigma and overall saffron production [146]. *Bacillus thuringiensis* DC1, *B. megaterium* VC3 and *B. amyloliquefaciens* DC8, isolated from saffron plant, have been characterized based on multiple plant growth promoting traits along with decrease in corm rot incidence [148]. Using a cultivation-independent 16S rRNA gene-targeted metagenomic approach, it has been observed that the cormosphere of saffron is dominated by *Pantoea,* whereas the rhizosphere is dominated by *Pseudomonas* and bulk soil by uncultivable *Acidobacteria* [139].

### 5.7. Integrated Disease Management

Due to inconsistent performance of biological control agents and as their efficacy is markedly affected by the abiotic factors, an integrated disease management (IDM) module, which is based on harmonized integration of compatible production and protection methods, is being advocated against the management of soil-borne diseases. IDM strategies have resulted in the enhanced germination (76.79%), reduced disease incidence (60.23%) and increased yield of saffron (62.62%), as compared to the farmers' practices [147]. Fusarium yellows of gladiolus has been managed by the integration of fungicides and biocontrol agents [148]. Demonstrations on IDM technology have been conducted under farmers' participatory approach (FPP) mode, wherein scientific guidance along with chemicals were provided to the beneficiaries, from sowing until harvesting (Figure 4). Due to the adoption of IDM in the farmers' fields, decrease in the corm rot of saffron was recorded and the productivity of saffron increased up to 3.96 kg ha$^{-1}$ [149,150].

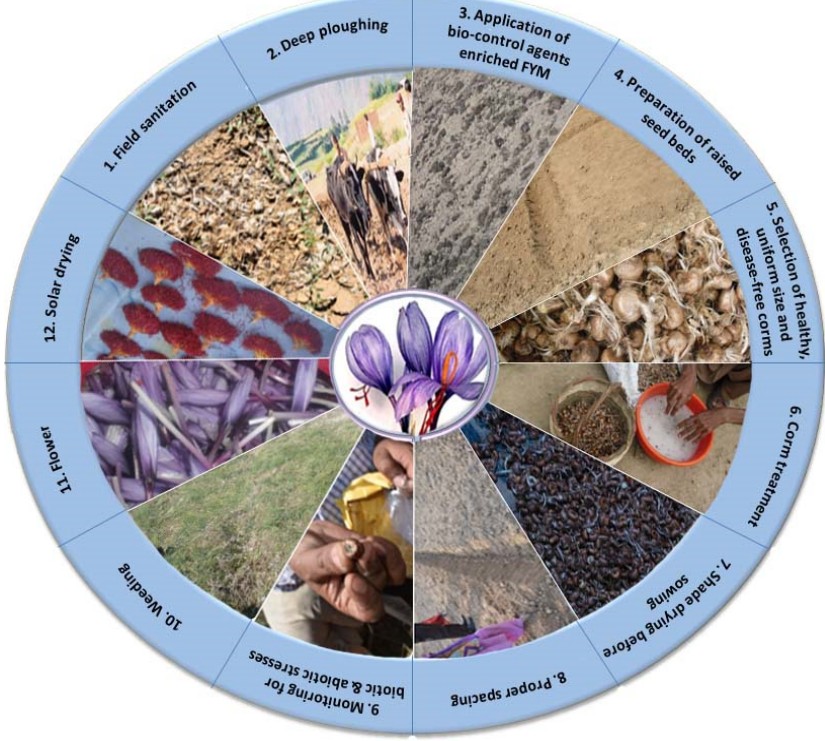

**Figure 4.** Integrated disease management (IDM) module for corm rot of saffron.

## 6. Summary and Future Thrust

In order to meet the extensive demand of saffron, because of its pharmacological potential and scarce production, efforts to explore the potential of developing resistant genotypes which are effective against corm rot complex represent an urgent need. Advanced methodologies of molecular breeding can be utilized for making the resistant genotypes available, which have higher productivity and resistance against *Fusarium oxysporum*. Due to the timely non-availability of seed corms to the farmers, developing a large quantity of planting material through tissue-culture is another way of ensuring healthy and timely supply of the corms. This will ensure horizontal and vertical expansion of the area under saffron cultivation.

Special attention is required for the biosecurity measures, which will help in the planting of disease-free materials in the fields and thus restrict the dissemination of disease within the area or across the state, country and continent. Periodical monitoring against the disease using diagnostic tools accelerates the rapid detection of the pathogens in corm and soil, that helps in recording the first appearance of the disease in the field and can avoid its further spread. This along with disease forecasting helps in coming up with an economically viable disease management module.

Improved cultural practices with the incorporation of integrated nutrient management tactics need to be standardized so that issues related to the soil health are properly addressed. Modifications of the existing traditional cultural practices in a specific agroclimatic condition are urgently required for the better management of the disease. Promoting the application of organic amendments, bio-fertilizers and bio-control agents, due to their persistence in the soil, not only reduces the pathogen inoculum but also stimulates the saffron growth and yield.

**Author Contributions:** Conceptualization, V.G. and V.K.R.; methodology, V.G., A.S. and P.K.R.; formal analysis, V.G.; investigation, S.K.G., B.S. and S.K.S. (Satish Kumar Sharma); resources, S.K.S. (Santosh Kumar Singh); R.H., D.K. and S.P. data curation, V.G. writing—original draft preparation, V.G.; writing—review and editing, V.G. and V.P.; visualization, V.G. and R.S.; supervision, V.G.; project administration, V.G. All authors have read and agreed to the published version of the manuscript.

**Funding:** This research received no external funding.

**Institutional Review Board Statement:** Not applicable.

**Informed Consent Statement:** Not applicable.

**Data Availability Statement:** Not applicable.

**Conflicts of Interest:** The authors declare no conflict of interest.

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
