# Peer review of "Corm Rot of Saffron: Epidemiology and Management"

_agronomy, doi:10.3390/agronomy11020339_

Round 1

Reviewer 1 Report

Dear Editor and Authors,

first of all I want to apologize for the delay in reviewing the present manuscript.

The present paper entitled "Corm rot of saffron: Epidemiology and Management" is focused in reviewing the current knowledge of corm rot of saffron. The topic is very valuable considering the economic importance of saffron and the need of reducing crop losses due to corm rot. Moreover, the scientific community could take advantages from this review, addressing in new researches the knowledge gaps on corm rot of saffron and its management. 

However, in my opinion the paper needs to be improved in different aspects. First of all, I suggest to focus on the main causal agents of corm rot and enter in more details about their epidemiology. Considering the title, epidemiology should be one of the key topic of this paper, and so, a consistent improvement about this section is needed. An other important aspect, should be to highlight the lack of knowledge in the epidemiology of some pathogens (e.g. for which extended studies are still needed?). After that, some improvements of the management paragraph are needed too.

Please, find more specific comments in the attached file.

Best regards 

Author Response

Dear Sir,

Responses to the reviewers' comments

Reviewer 1:

I would suggest to describe symptoms based on the causal agent, mostly because in Figure 3 you reported different genera causing very different symptoms on corm.

The symptoms caused by causal agent is  mentioned in the text and highlighted with blue colour.

Deleted the word (October-November).

Here you reported many microorganisms associated mainly with the cormosphere or endophytes. So, I suggest to focus mainly on organisms for which there is evidence for causing corm rot.

Section is divided into pathogens and myco-flora associated with corm rot. Repeated line (272) deleted. Underline removed.

In my opinion, this section should be improved.

Only limited work has been carried out related to epidemiology of the corm rot of saffron. However, additional information (highlighted) added in the disease cycle and epidemiology section.  

I would move this in the previous paragraph.

Necessary information has been shifted in the suitable section.

I suggest you to clearly separate the topics including in this paragraph, creating different subparagraphs (e.g., storage, sowing, etc.)

We thank reviewer for the right suggestion. But only scanty information is under the cultural practices so no changes are made.

Reviewer 2 Report

The paper is very interesting. It represents a good overview and gives useful information for readers of Agronomy.
The corm rot of saffron was addressed on various aspects.
I would like suggest to authors to improve Figure 3. It shows some slight blurry photos and it is not focusing on the different corm rots, so it is dispersive, while the title of the work makes this expectation 

Author Response

Dear Sir,

I would like suggest to authors to improve Figure 3. It shows some slight blurry photos and it is not focusing on the different corm rots.

Resolution of Figure 3 improve and added. It only depicts the different stages of corm rot.

regards

Reviewer 3 Report

The disease infestition in Iran wchich produces about 90% of world saffron production as reported is not significant (2 ref.).  I suggest some explanation for this, but in general very informative, well reviewed and written.

Author Response

Dear Sir,

The disease infestation in Iran which produces about 90% of world saffron production as reported is not significant (2 ref.).

Information was based on only published literature.

regards